# Study of Nuclear Reactions in Therapy of Tumors with Proton Beams

**DOI:** 10.3390/ijms241713400

**Published:** 2023-08-29

**Authors:** Maxim Azarkin, Martin Kirakosyan, Vladimir Ryabov

**Affiliations:** P. N. Lebedev Physical Institute, 119991 Moscow, Russia; kirakosyanmr@lebedev.ru (M.K.); ryabov@lebedev.ru (V.R.)

**Keywords:** proton therapy of cancer, simulation of proton therapy, nuclear reactions in proton therapy, proton therapy with radiosensitizers, Geant4, nanoparticles in proton therapy

## Abstract

This paper presents an assessment of nuclear reaction yields of protons, α-particles, and neutrons in human tissue-equivalentmaterial in proton therapy using a simulation with Geant 4. In this study, we also check an enhancement of nuclear reactions due to the presence of Bi, Au, ${}^{11}$B, and ${}^{10}$B radiosensitizer nanoparticles. We demonstrate that a proton beam induces a noticeable amount of nuclear reactions in the tissue. Nevertheless, the enhancement of nuclear reaction products due to radiosensitizer nanoparticles is found to be negligible.

## 1. Introduction

The treatment of cancer with proton beam therapy allows better targeting of the tumor over gamma-ray therapy. The advantage is caused by a significant increase in deposited energy at the very end of the proton track, i.e., a well-known Bragg peak. Similarly to gamma rays, protons mainly lose their energy through the ionization of the medium. Moreover, protons can cause nuclear reactions, leading to the production of neutrons, gamma rays, α-particles, unstable isotopes, and other products of nuclear fission. The role of nuclear processes in proton therapy is still not well understood.

Despite a better targeting of tumors by proton beam compared to gamma rays, protons inflict noticeable damage to healthy tissue on the way to the tumor. Therefore, medical physicists have been seeking a way to increase damage to the tumor while simultaneously keeping irradiation of healthy tissues at an acceptable level. Thus, nearly 40 years ago it was found that high-*Z* elements can be used as radiosensitizers, enhancing the effective dose in radiotherapy [1,2]. The effect is based on huge electromagnetic cross-sections at low energies that lead to an increase in the local dose deposition. Both in vitro and in vivo studies show that the presence of nanoparticles of high-*Z* materials in cancerous tissue with concentrations of ∼100 ppm (or even less) [3] significantly improves the therapeutic effect of radiotherapy. Initially, the effect was studied for traditional gamma therapy. However, a number of relatively recent studies show that the increase in effective dose can also be observed in proton therapy [4,5,6,7,8,9,10]. The effect may be present in proton therapy due to both the electromagnetic interactions of beam protons and secondary electrons, photons with nanoparticles, and nuclear interactions of protons with the material of nanoparticles. Moreover, nuclear interactions are of special interest since they can result in the production of α-particles which have high linear energy transfer (LET) and, therefore, relative biological effectiveness (RBE).

In order to better understand the underlying mechanisms, simulations of proton beam interaction with tissue are needed. There are a number of simulation studies of interactions of gamma rays and proton beams with tissue-like (mostly water) systems with incorporated nanoparticles consisting of various materials. A significant increase in dose in the vicinity of a nanoparticle was found for both gamma-ray and proton therapy [11,12,13].

In [13], it was shown that there is a local microscopic increase in fluences for both α-particles and neutrons in the vicinity of ${}^{157}\mathrm{Gd}$ nanoparticles. Compared to other high-Z elements, gadolinium is of special interest because of the huge cross-section of thermal neutron capture (250,000 barn). Furthermore, the role of nuclear reactions in proton therapy draws a lot of attention in the context of the usage of ${}^{11}B$ as a radiosensitizer [9]. A number of studies suggest that experimentally observed effectiveness of boron nanoparticles [9] may be explained by a relatively high cross-section (up to ≈1200 mb at $T_{p}$ = 675 keV) of boron–proton fusion reaction that results in the production of three α-particles: (1)$$p+{}^{11}B\to 3\alpha +8.7\;\mathrm{MeV}\;(\sigma \approx 1200\;\mathrm{mb})$$

It should be noted that the process has a maximum cross-section at relatively low energies which corresponds to energies at the Bragg peak. Repulsion between protons and ions due to Coulomb force prohibits protons from getting close enough to the nucleus for interaction. The resonant nature of proton–boron interaction makes it possible to take place below the Coulomb barrier. This cross-section is well studied since it is considered to be a candidate for an aneutronic nuclear fusion. There is a significant amount of studies that use Monte Carlo simulations to estimate the contribution of proton–boron fusion to the increase in a local production of α-particle in proton therapy. The conclusions are controversial. Some studies suggest this contribution is significant [14], others indicate that the enhancement of α-particle production is negligible [15,16].

The above-mentioned high-*Z* radiosensitizers have cross-sections of inelastic nuclear reactions of the same order as proton–boron fusion ($\sigma_{in}∼$ 1000 mb). For instance, widely used bismuth and gold radiosensitizers experience following nuclear reactions:(2)$$p+{}^{209}Bi\to \left[{}^{210-x}Po\right]+xn,\;x=1,2\ldots \;\left(\sigma_{xn}\approx \;1000-1200\;\mathrm{mb}\right)$$
(3)$$p+{}^{197}Au\to \left[{}^{198-x}Hg\right]+xn,\;x=0,1,2\ldots \;\left(\sigma_{xn}\approx \;1000-1200\;\mathrm{mb}\right)$$

A significant fraction of produced Po and Hg isotopes decay with the emission of α-particles. It is also quite probable that excited Bi and Au nuclei can directly yield α-particles. This opens a question of the significance of these nuclear reactions in proton therapy. Such reactions require higher energies; nevertheless, a significant fraction of protons is expected to have energies high enough to trigger nuclear processes. This is due to two main factors:Cancer tumors have sizes (order of ∼1 cm) that mean that protons, reaching the rear side of the tumor, have excessive energy at the front side.The probabilistic nature of proton interactions with the matter results in a noticeable spread of the energy spectrum of protons at the Bragg peak. Moreover, the longer the proton’s path to the target (tumor) the larger the energy spread.

For that reason, we will study the role of nuclear reactions with the gold and bismuth radiosensitizers.

Another important subject related to nuclear processes in proton therapy is the impact of neutrons [13,17]. Neutrons have a large RBE, but, unfortunately, they easily drift away to healthy tissues. For that reason, their role should be carefully assessed both in the presence of radiosensitizers and without one. It is worth mentioning that neutrons rapidly slow down (see Section 2), reaching very low energies (down to thermal ones) due to collisions with hydrogen atoms. This opens the possibility of using elements with a large neutron capture cross-section as radiosensitizers. For instance, it was suggested to use such radiosensitizers to increase the absorption of neutrons in tumors and decrease the exposure of healthy tissue to neutron radiation [13] (in this particular research, ${}^{157}$Gd was considered). For this reason, we add ${}^{10}$B to the list of radiosensitizers to test in this study since the boron isotope ${}^{10}$B has been successfully used for neutron capture therapy in clinical practice for several decades [18,19,20], and has a very high neutron capture cross-section value:(4)$$n+{}^{10}B\to {\left[{}^{11}B\right]}^{*}\to \alpha +{}^{7}Li+2.38\;\mathrm{MeV}\;\left(\sigma \approx 3837\;b\right)$$

As far as authors are concerned, currently, there is a gap in all-around simulations of nuclear reactions in proton therapy with heavy nanoparticles; however, the importance of simulating hadronic processes in proton therapy was emphasized in [11]. Simulation studies mostly look into microscopic systems where protons or gamma rays directly hit nanoparticles. There is a lack of full simulations that take into account the flux of secondary particles. One of the purposes of the presented work is to fill this gap. We simulate the proton beam interaction with tissue containing boron isotopes, gold, and bismuth nanoparticles using Monte Carlo simulation of a particle passage through matter with Geant 4 [21]. We will check whether admixture of ${}^{11}B$, ${}^{10}B$, Bi, or Au particles to the tumor can enhance nuclear reaction yields. Fluences, energy spectra, and densities of produced particles will be measured in the simulation of the interaction of proton beams with a tissue-like material using Geant 4 toolkit [21].

In Section 3 of this paper, we discuss the methods we use for our study; in Section 2 the results of the simulation are presented; Section 4 is left for conclusions.

## 2. Results and Discussion

In the context of the presented work for the assessment of hadronic interactions in proton therapy, it is instructive to start with a simplistic geometric representation which gives a clear physics picture of the interaction profile. At the end of this section, we will present results for a quite realistic human phantom. Thus, we start with the simulation of the interaction of proton beams with the system represented by the cube with a side of 100 mm filled with a simplistic soft tissue-like material, which can be accessed in the Geant 4.11 calling class G4HumanPhantomMaterial with the argument “soft_tissue”. The material consists of a simple mixture of elements that comprise human soft tissue and have a density of human soft tissue. For the sake of brevity, we will use the word “tissue” instead of “tissue-like material” further in the text. The proton beam crosses the cube perpendicularly to one of its faces. The layer located on the depth between 50 mm and 60 mm represents a malignant tumor. A layout of the simulated system is shown in Figure 1. In this study, we investigate nuclear reactions for four cases: uniform cube consisting of pure tissue-like material, and four cases of tumor layers with incorporated Bi, Au, ${}^{10}$B, or ${}^{11}$B particles. In this study, atoms of the tissue and radiosensitizer are uniformly distributed. This assumption well describes the tissue and is efficiently applicable to radiosensitizers in the context of the study of nuclear reactions. The latter can be justified by the fact that the absorption length of α-particles, photons, and neutrons produced in nuclear reactions of beam protons and nuclei of radiosensitizers is way larger than the typical distance between radiosensitizer nanoparticles. For instance, a typical therapeutic concentration of 100 ppm for gold nanoparticles with a size of 20 nm gives the average distance of ≈0.5 μm, whereas the absorption length of produced α-particles is at least 50 μm. A low yield of nuclear reactions per 1 nanoparticle (≪1) also makes it nearly impossible to produce microscopic hot spots in the vicinity of an excited nanoparticle. These two facts make the size of radiosensitizer particles insignificant for the study of nuclear reactions. This is in striking contrast with the electromagnetic case where excitations of atoms of radiosensitizer particles by the flux of secondary electrons and photons result in a huge increase in electron production and energy deposit in the vicinity of a nanoparticle.

In this study, we assume that radiosensitizers do not migrate out of the tumor. This assumption may be a significant simplification in real medical applications since some radioactive isotopes produced in nuclear reactions have noticeable decay times. For instance, a proton–bismuth nuclear reaction produces isotopes of Po which may have a half-life time of up to a few days (and more). However, half-life times of most Hg isotopes produced in proton–gold nuclear reactions do not exceed a fraction of a second.

In this study, we are focused on the production of neutrons and α-particles since they have high RBE and are abundantly produced in the nuclear reaction of protons with tissue and radiosensitizers. It should be noted here that we do not discuss the production of gamma rays in nuclear reactions since it contributes <0.1% to the total deposited dose and has a low LET and therefore has a negligible biological significance in proton therapy.

In order to provide a clear sharp picture of nuclear processes, the concentrations of these radiosensitizing metals are set to 1000 ppm (typically, in vivo and in vitro studies use concentrations of radiosensitizers from 10 to 100 ppm). We also perform a simulation for two scenarios of the beam proton energy ($T_{\mathrm{beam}}$), namely $T_{\mathrm{beam}}$ = 87 MeV and 95 MeV. The first one is adjusted so that the Bragg peak is close to the rear side of the tumor layer. In the second scenario, the proton energy is moderately increased by 8 MeV to check if nuclear reaction yields can be significantly enhanced, in particular in the tumor slice with incorporated radiosensitizer particles. Figure 2 justifies the choice of beam energies by showing the energy loss of beam protons as a function of depth in these scenarios.

The picture of the interaction of proton beams with tissue is generally characterized by fluences of protons, produced neutrons, and α-particles. They are shown for $T_{\mathrm{beam}}$ = 87 MeV and 95 MeV in Figure 3. One can see that the lower beam energy provides absorption of most of the beam protons at the depth of 60 mm, whereas for higher beam energy, protons are absorbed at a depth of ∼70 mm. Neutron fluences are shown for all neutrons (i.e., without kinematic cuts) and separately for thermal neutrons (T< 0.5 eV). Most of the neutrons experience multiple scattering that is reflected in a bumpy shape of the dependence of fluence on depth. As mentioned in the introduction, scatterings on hydrogen atoms lead to a significant fraction of slowed-down neutrons at thermal energies. It is worth mentioning that neutron fluences may depend on the geometry of the simulated system.

Inelastic nuclear reactions between proton and nuclei of the tissue require noticeable energy (order of 10 MeV). This is also true for Au and Bi radiosensitizers. To justify the possibility of nuclear reactions in the tumor slice, which is located at the end of the beam proton track, we demonstrate in Figure 4 kinetic-energy spectra of beam protons crossing the front and rear plane of the tumor for $T_{beam}=$ 87 MeV and 95 MeV. In the first case, 93% of initial beam protons reach the front plane of the tumor slice having T≈31 MeV, and only 12% of beam protons reach the rear plane with T≈4 MeV. This means that nuclear reactions mostly take place in the front part of the tumor slice, where an average *T* of beam protons is higher than the typical binding energy (up to 8 MeV) of nucleons in nuclei. For the second scenario of beam energy, 94% of initial beam protons hit the front plane of the tumor with T≈46 MeV, 92% of initial beam protons leave the tumor with T≈31 MeV. Nuclear reactions are more prevalent in this scenario, which is reflected in an enhancement of fluences of produced neutrons at a depth >50 mm for $T_{\mathrm{beam}}$ = 95 MeV in comparison to the case of $T_{\mathrm{beam}}$ = 87 MeV. It is instructive to note that protons after a depth of >65 mm (75 mm) for $T_{\mathrm{beam}}$ = 87 MeV (95 MeV) are produced in secondary nuclear reactions between neutrons and tissue nuclei. The yield of such nuclear reactions is rather low.

In order to understand the underlying mechanisms of the therapeutic effects of treatment with proton beams, we estimate the number of α-particles and neutrons produced in nuclear reactions as a function of depth per 1 beam proton. These functions are shown in Figure 5 and Figure 6 for $T_{\mathrm{beam}}$ = 87 MeV and 95 MeV, respectively. Four cases of radiosensitizer materials are simulated: bismuth, gold, boron-10, and boron-11. The common feature of all these histograms is the fact that the production rate of α-particles has a prominent peak at a depth that is 4–5 mm before the Bragg peak. The production rate at the peak is 3 times higher in comparison to the one at the entry point of the proton beam. The observed picture is consistent with experimental data since the cross-section of α-particle production in nuclear reactions between protons and light nuclei such as oxygen, carbon, and nitrogen has a wide peak for *T*= 10–20 MeV [22]. As we can see from Figure 5 and Figure 6, Bi, Au, and ${}^{11}$B particles do not bring any visible enhancement of production of α-particles, whereas the activation of ${}^{10}$B by secondary thermal neutrons results in the enhancement of α-particle production by 5%. It should be noted that boron-10 also produces ${}^{7}$Li nuclei (see Reaction (Equation 4)) that will contribute to the biological effectiveness of ${}^{10}$B. ${}^{7}$Li has high LET and a short absorption path (around 5 μm). The production of neutrons decreases with depth, which is also expected. However, there is a clear increase (30–70%) of neutron production in the tumor slice if it is enriched with Au, Bi, and ${}^{10}$B particles. One should remember that all of the above results are obtained for a concentration of 1000 ppm, which is far higher than typical therapeutic ones.

Another important characteristic for understanding interactions of produced particles with the tissue is their kinetic energy at the production point. Figure 7 shows kinetic energy distributions for produced α-particles and neutrons in the tumor slice under a proton beam with an energy of 87 MeV. One can see that the distributions are quite broad. Most α-particles are produced with energies up to ≈15 MeV (6 MeV on average) that allows them to travel a distance up to ≈300μm (50 μm) from the production point. The mean energy of neutrons at the production point is 2.7 MeV. One should mention that the difference in energy spectra of α-particles and neutrons produced in the tumor slice of pure tissue and the tissue with radiosensitizer particles is negligible.

It is instructive to assess the role of nuclear reactions in proton therapy, namely the amount of produced α-particles per dose of 1 Gy per one cell. It can be concluded from Figure 4 that a proton loses about 25 MeV in the last centimeter before termalization. Thus, the dose 1 Gy deposited in 1 cm${}^{3}$ of tissue corresponds to irradiation by $\approx 2.5\times 10^{11}$ protons. The average yield of α-particles per 1 proton and 1 per 1 mm of the path in the tumor slice is estimated to be $1.4\times 10^{-3}$ (see Figure 5). These values give $\approx 2.5\times 10^{11}$ ($\frac{\mathrm{protons}}{{\mathrm{cm}}^{3}\cdot \mathrm{Gy}}$) $1.4\times 10^{-3}$($\frac{\alpha -\mathrm{particle}}{\mathrm{mm}\cdot \mathrm{proton}}$)10 (mm) = $3.5\times 10^{9}$ ($\frac{\alpha -\mathrm{particle}}{\mathrm{Gy}\cdot {\mathrm{cm}}^{3}}$). For instance, 1 cm${}^{3}$ of tissue contains $\approx 3.7\times 10^{9}$ cells with a size of 30 μm. Thus, a dose of 1 Gy delivered with a proton beam gives ≈1α-particle per cell in the tumor slice. One should remember that the production rate of α-particles in the preceding tissue is 2 times lower than in the tumor slice.

It should be noted that nuclear reaction yields in proton therapy practice may differ from the values obtained in the above-presented simulation of a simplistic system. Spreading out of Bragg peak and heterogeneity of irradiated parts of human body smear proton energy spectrum in the tumor volume. This may affect nuclear reaction yields. For example, human and animal bones have higher density and their composition (60% inorganic hydroxyapatite, 10% water, and 30% organic components) differs from the composition of soft tissues. Moreover, the configuration of an irradiated system as a whole determines the process of neutron reflection and neutron thermalization. Therefore, for the sake of integrity, we performed a simulation of proton therapy of a tumor inside a human chest using one of the realistic human phantoms ICRP145 [23] recently implemented in Geant 4.11. Namely, we used an adult male phantom obtained with CT imaging of real persons. The graphical representation of the human phantom is given in Figure 8. The tumor is located behind the breastbone at a depth of ≈8 cm (i.e., nearly in the middle of the chest) and is 1 cm in diameter. It consists of soft tissue. Results of the simulation on yields of α-particles and neutrons averaged over the tumor volume are given in Table 1. The yields are estimated for a bare tumor and a tumor with radiosensitizer particles inside. The differences with respect to the simplistic case are minor for most observables. However, one should note that the contribution of ${}^{10}$B to the α-particle production increases by a factor of ≈1.5. This is expected since a larger system better reflects neutrons back and, therefore, the fraction of thermal neutrons is increased.

The robustness of obtained results was verified by using other general purpose physics lists. Thus, we performed our simulations using QGSP_BERT_HP, QGSP_BIC. No significant quantitative differences were observed for most particles in comparison to QBBC except for neutron fluence in QGSP_BERT_HP, because of the usage of G4NeutronHP model for neutron transport in this physics list. However, the observed 20% difference is of no significance to the conclusions of this study.

## 3. Methods and Materials

We perform the study using Geant 4 (version 4.11.1.1), an open-source package for Monte Carlo simulation of particle propagation in matter [21]. Nowadays, Geant 4 is widely used in various fields of physics, from high energy and cosmo physics to medicine [24]. In the Geant 4 approach, the user specifies the configuration of the studied system, processes, and physical models for the interaction of beam particles with given media. After the configuration is specified Geant 4 Monte Carlo algorithms perform the simulation.

The aim of this study is to investigate the contribution of nuclear interactions to the high-LET particle production in proton therapy, which is why we will give a more detailed description of the hadron physics that is being used in simulations. We will not discuss electromagnetic (EM) interactions, since it goes beyond the scope of this paper. However, it is important to note that the standard Geant 4 subroutines for EM processes well describe the Bragg peak in various media.

Geant 4 is an object-oriented tool for Monte Carlo simulations of particle transport in matter. The object-oriented nature of Geant 4 allows one to use models for different physics processes at various energy ranges. During the years of development, many models have been implemented for different purposes. Usually, for the simulation, certain sets of pre-packaged physics models are used, also called physics lists. Among the most widely used lists in high-energy physics and medicine is QBBC [25,26], which is also utilized in this study. This physics list implements standard Geant 4 electromagnetic models, QGSP (Quark Gluon Strings, string base model + Precompound for nuclear de-excitation) for high-energy hadron interactions (>8 GeV), binary cascade for protons and neutrons [27], Bertini cascade [28] for mesons and other baryons, CHIPS (CHiral Invariant Phase Space) models for low energy capture of π, *K*, antiprotons, and μ and low-energy photonuclear reactions.

Within the context of the current study, the interesting part of QBBC is the binary nuclear cascade model (also referred to as BIC), which simulates interactions of protons (and other hadrons) with nuclei of the matter. Let us briefly describe here the key features of this model. The model describes the nucleus as a three-dimensional spherical isotropic object, nucleons inside the nucleus being spatially distributed by either Gaussian distribution (for atomic number A<17) with a variance $R\propto A^{1/3}$, or by Wood–Saxon form (for heavy nuclei):(5)$$\rho \left(r\right)=\frac{\rho_{0}}{1+\exp\left(\frac{r-R}{a}\right)},$$
where *R* and $r_{0}$ depend on *A*, $R=r_{0}\cdot A^{1/3}$, and $r_{0}=1.16\left(1-1.16A^{-2/3}\right)$ fm; a=0.545 fm. Additional conditions demand that the minimal distance between nucleons should be >0.8 fm. Nucleons inside nuclei carry momentum uniformly distributed from 0 to Fermi radius $p_{F}∼\rho {\left(r\right)}^{1/3}$ such that the sum of all momenta should be 0.

The interactions of hadrons (or nuclei) with the nuclei are simulated by means of binary interactions of hadrons and secondaries with nucleons of the nucleus. Collective interactions inside the nucleus are taken into account by the transport in intranuclear potential $V\left(r\right)=p_{F}^{2}/2$ m (the mass of proton or neutron) for baryons. For pions, a slightly more complicated parametrization is being used (see [27] for details). Binary baryon or meson interactions are either derived from experimental data, for elastic scattering, or by t- or s-channel resonance excitations. Decay-branching ratios are taken from the PDG (Particle Data Group [29]). Pauli blocking is used for all interactions, i.e., the reaction is allowed only when final states have momentum $>p_{F}$. The Coulomb barrier is included in a binary cascade model in Geant 4 for charged particles.

The binary cascade model proved to adequately describe experimental data for proton–nuclei interactions at different energies. For instance, in [27], it was shown that for 800 MeV p-Pb interactions, the binary cascade model reproduces experimental data of neutron yield within errors. In the same paper, it was demonstrated that the model agrees with the experimental data of neutron production for carbon–carbon interactions at energies of 290 MeV/nucleon. In another study [30] it is demonstrated that BIC describes the experimental data for 65 MeV proton–ion interactions.

To simulate the de-excitation of the formed nucleusto the equilibrium after the kinetic stage, the precompound model [31,32] is implemented in QBBC. The model uses an atomic number *A* of the nucleus, its charge *Z*, four-momentum *P*, excitation energy *U* (defined from conservation laws at the cascade stage), and number of excitations *n*, which is a sum between the number of exciton particles *p* (i.e., the number of captured nucleons during the cascade stage) and holes *h* (number of nucleons produces in intranuclear collisions). The system relaxation to equilibrium and emission of nucleons and complex fragments (deuterium, tritium, and helium are considered in Geant 4) is described by this model.

The emission of fragments from the equilibrated nucleus (which means excitation energy is not localized and is shared between many nucleons) and excited nuclei are described by the evaporation model [32]. For the high-Z nuclei, nuclear fission model is implemented (A>65) in Geant 4, whereas for light nuclei (A<17, Z<9), the Fermi model is used to describe the statistical breakup [33]. For high excitation energies, a statistical breakup is described by the multifragmentational model [33].

Subsequent decays of produced isotopes with emissions of α, $\beta^{+}$, $\beta^{-}$, and γ are handled by data-driven tables which are encoded in the physics module G4RadioactiveDecay. The decay of produced isotopes is especially important for high-*Z* nuclei in the context of this study, and therefore it is switched on in our simulation.

Dedicated neutron transport models simulate interactions of neutrons in media at low energies (up to ∼20 MeV). The model splits neutron interactions into four parts. Namely, elastic scattering, neutron capture, fission, and inelastic scattering. In standard Geant 4 (implemented also in QBBC physics list), inclusive cross-sections are taken from the library ENDF/B-VI [34] and point-wise approximation is used. Differential cross-sections of each part are also tabulated using ENDF/B-VI data. A more elaborate high-precision model (G4NeutronHP) for neutrons is also being used in Geant 4 (not in QBBC physics list); however, the use of G4NeuronHP is not significant for our study (see Section 2).

## 4. Conclusions

In this paper, we presented a study of nuclear reactions in proton therapy using MC simulations with Geant 4. We simulated interactions of protons with nuclei of soft human tissue. Furthermore, we examined whether such widely studied radiosensitizers as Au, Bi, ${}^{10}$B, and ${}^{11}$B can be useful for the enhancement of the yield of nuclear reactions. Of special interest in the context of proton therapy is the production of α-particles which have high energy transfer (up to 100 keV/μm) to surrounding medium and an absorption length comparable with cell size.

The most abundant products of nuclear reactions induced by proton beams in soft human tissue are α-particles and neutrons. The production rates of these particles are 0.6–0.8 $\times 10^{-3}{\mathrm{mm}}^{-1}$ per 1 beam proton with energy above ≈35 MeV. In the pre-Bragg peak area, the production rate of α-particles increases up to 1.8 × 10${}^{-3}$
${\mathrm{mm}}^{-1}$, whereas for neutrons, it drops down to 0.1–0.2 × 10${}^{-3}$
${\mathrm{mm}}^{-1}$. Unlike α-particles, produced neutrons have an absorption length of a few orders higher. However, a significant fraction of them is thermalized. Thus, a fluence of thermal neutrons is up to $10^{-3}$ per 1 beam proton. That is the reason we checked whether the injection of ${}^{10}$B, which is used as an absorber in neutron capture therapy, can be useful in proton therapy. It should be noted that the interaction of proton beams with tissue produces a noticeable amount of α-particles and neutrons. Given the fact that they have high RBE, their biological effects (e.g., DNA breaking) in proton therapy should be carefully assessed.

It is argued that radiosensitizers may significantly enhance the production of α-particles [13]. Our study demonstrates that the enhancement is insignificant or even negligible. We simulated four scenarios of Au, Bi, ${}^{10}$B, and ${}^{11}$B particles delivered into the tumor area with a concentration of 1000 ppm. The first and second ones do not show any statistically significant differences with respect to the pure tissue. The third one leads to an increase of 5–10% that can be explained by the extremely high cross-section of neutron capture by ${}^{10}$B (σ≈3800 b). The contribution of ${}^{11}$B fission to the total production of α-particles in tissue is negligible as well. Taking all observations together, we conclude that none of the above-mentioned radiosensitizers can bring meaningful enhancement to α-particle production. The simulation of the proton therapy with human phantom ICRP145 demonstrated that, as expected, the conclusions do not change in more realistic scenarios. One should also remember that radiosensitizer concentrations for demonstrative purposes were set to 1000 ppm, which is far above typical therapeutic values.

The effect of radiosensitization with nanoparticles observed in proton therapy appears to be due to the local microscopic increase in dose delivered by secondary electrons and photons emitted by excited atoms of radiosensitizer nanoparticles. This point of view is generally accepted for now [3,11]; nevertheless, the majority of such estimates are still poorly elaborated. Let us stress that, contrary to nuclear reactions, electromagnetic processes have huge cross-sections at keV energies: the cross-sections of photoelectric processes for high-*Z* elements (note, that it is proportional to $Z^{5}$ at sufficiently low energies) reach megabarns, and produced electrons and γ have short absorption lengths in tissue. For example, MC simulations show that during the irradiation of Au nanoparticles in water with 50 kvp X-rays, 65% of the energy of $e^{-}$ and γ emitted by Au atoms was deposited in the 100 nm region outside the nanoparticle [35]. Thus, since a significant part of the energy is absorbed at distances much smaller than the average distance between nanoparticles, the full simulation of the system on a micro/nanoscale is needed for the correct assessment of the effect of electromagnetic interactions in proton therapy.

## Figures and Tables

**Figure 1 ijms-24-13400-f001:**
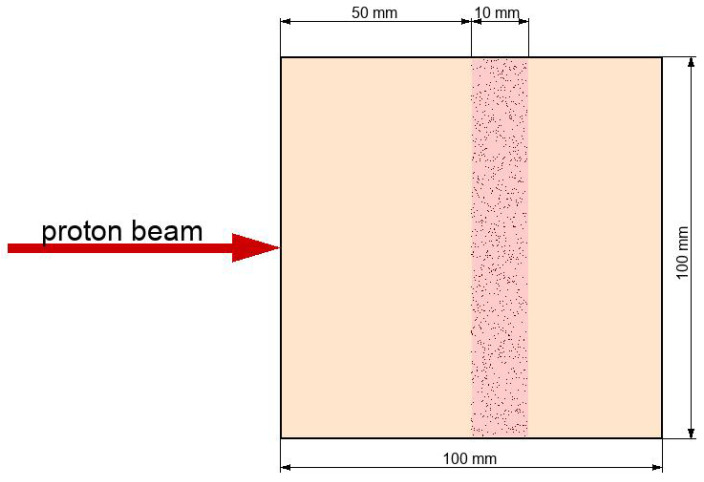
Layout of the simulation setup. The setup represents a cube with a size of 100mm×100mm×100mm which encompasses a tumor slice spanning in depth from 50 mm to 60 mm. The tumor slice is highlighted by pink and in most simulation cases includes radiosensitizer particles. The proton beam direction is indicated with a red arrow.

**Figure 2 ijms-24-13400-f002:**
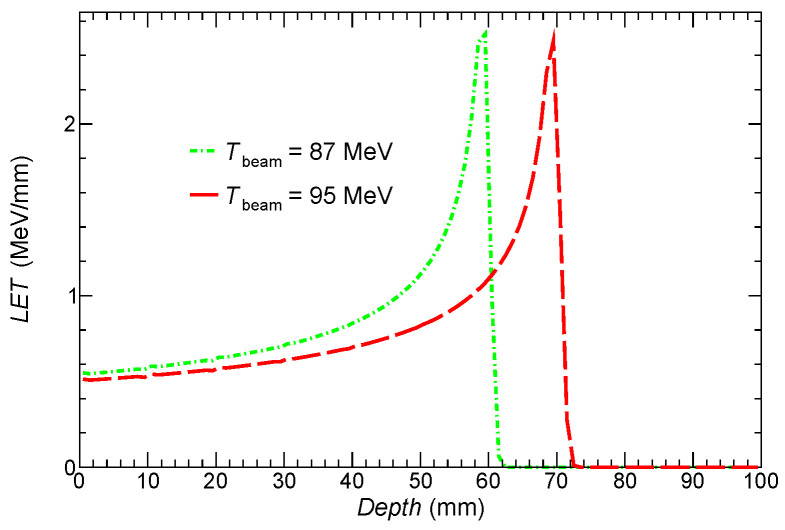
Linear energy transfer (LET) of beam protons as a function of depth of penetration for two different initial energy of beam protons.

**Figure 3 ijms-24-13400-f003:**
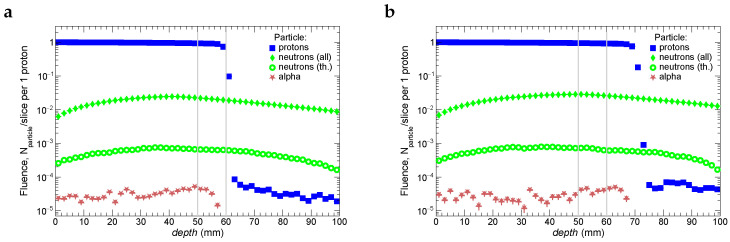
Fluences of protons, produced α-particles, and neutrons as a function of depth. The initial energies of beam protons are 87 MeV for the plot (**a**) and 95 MeV for the plot (**b**).

**Figure 4 ijms-24-13400-f004:**
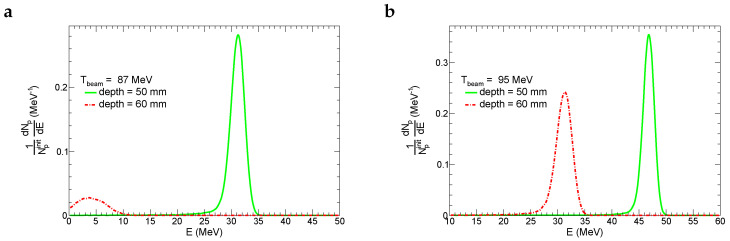
Energy distribution of survived beam protons ($N_{p}$) at the front and rear sides of the tumor layer. These distributions were normalized by the number of initial beam protons($N_{p}^{\mathrm{init}}$). Results are presented for two initial energies of beam protons ($T_{\mathrm{beam}}$): 87 MeV (**a**) and 95 MeV (**b**).

**Figure 5 ijms-24-13400-f005:**
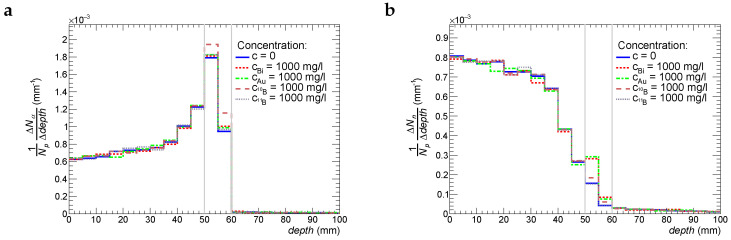
Production rate of α-particles (plot **a**) and neutrons (plot **b**) as a function of depth in tissue for the initial beam–proton energy of 87 MeV.

**Figure 6 ijms-24-13400-f006:**
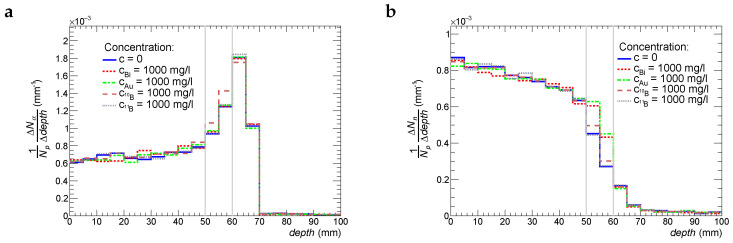
Production rate of α-particles (plot **a**) and neutrons (plot **b**) as a function of depth in tissue for the initial beam–proton energy of 95 MeV.

**Figure 7 ijms-24-13400-f007:**
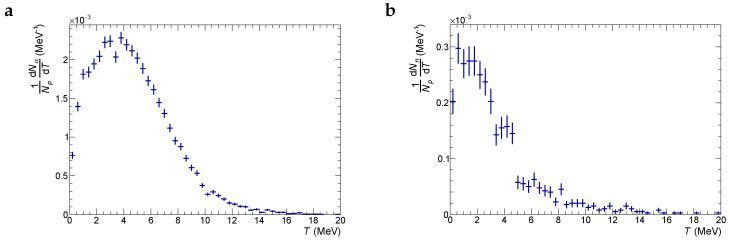
Spectra of kinetic energy of produced α-particles (plot **a**) and neutrons (plot **b**) in the tissue layer laying in depth range from 50 mm to 60 mm for the initial beam–proton energy of 87 MeV.

**Figure 8 ijms-24-13400-f008:**
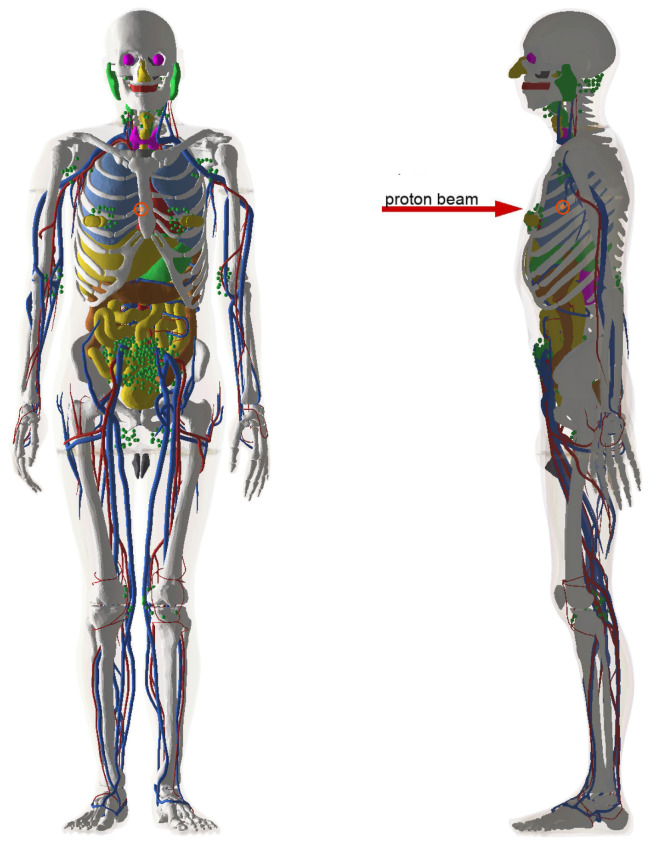
Adult male phantom ICRP145 [23] as implemented in Geant 4.11. The tumor is indicated with the orange marker. Proton beam direction is shown as a red arrow.

**Table 1 ijms-24-13400-t001:** Average yields of α-particles and neutrons inside the tumor volume per one beam proton.

Nanoparticle Material	Yield of α-Particles,	Yield of Neutrons,
	**$10^{-3}$ mm${}^{-1}$**	**$10^{-3}$ mm${}^{-1}$**
none	1.33	0.1
Au	1.36	0.17
Bi	1.35	0.17
${}^{10}$B	1.74	0.11
${}^{11}$B	1.37	0.1

## Data Availability

The code used for simulations presented in this study is available upon request to authors.

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
