# Peer review of "Study of Nuclear Reactions in Therapy of Tumors with Proton Beams"

_ijms, 2023, doi:10.3390/ijms241713400_

Round 1

Reviewer 1 Report

The manuscript concerns the valid problem of cancer therapy. In particular, it is devoted to the proton beams method. The Authors concentrate on enhancing the effectiveness of nuclear reactions when Bi, Au, ^{10}B, or^{11} B radiosensitizer nanoparticles are applied. They simulated how the presence of such particles can enhance the -particle production. To perform the simulations, they applied the binary nuclear cascade model for interactions of protons (and other hadrons) with nuclei of the matter. Technically, the simulations were performed using an open-source package GEANT 4 (version 4.11.1.1) for Monte Carlo simulations. The Authors have shown that the radiosensitizer particles enhance processes of the -particles only to a limited degree for ^{10}B  particles, and such enhancement does not exhibit for the three others. Moreover, the Conclusions section also comprehensively discusses the processes appearing during interactions with proton beams. 

The results, although not revolutionary, are sufficiently valid to be published. The manuscript is well written in general and is accessible to the Readers. The Authors also provide a list of references helpful in the studies of the topics presented in the article. Finally, it can be stated that the paper can be accepted in its present form.

Author Response

Dear referee,

Thank you for reviewing our paper and giving a positive response. 

Kind regards,
Martin, Maxim, Vladimir

Reviewer 2 Report

The team used the simulator GEANT 4 to study the effect of proton beam therapy against cancers such as nuclear reaction, effect of radiosensitizers, etc. The main intention is to enhance the proton beam treatment efficacy and reduce collateral damage to adjacent normal tissue. Here are my comments.

a)    How do the author justify the thickness of cube is 100mm, which is just 10cm? It could be meaningless to tumors in visceral organs because of the thickness.

b)    The team only used soft tissue like material for simulation. How about tumors that are covered by bones? for example, glioblastoma, in which proton ion beam therapy is one of the options. To improve the versatility of the simulator, the team should include various possibilities and demonstrate different outcomes.

c)    I believe it is insufficient to just provide all simulation results. The authors should consider using organoid models or mouse models to validate the outcomes.

Minor spelling check is required. 

Author Response

Dear referee,

Thank you for reviewing our paper. Your remarks were very valuable for us. 
The detailed in-line answer can be found in the attachment.

Kind regards,

Vladimir, Martin, Maxim

Reviewer 3 Report

The article "Study of nuclear reactions in the therapy of tumors with proton beams" presented to a friend concerns important issues for man directly related to his life and health. The reviewer sees the need to conduct research in this context. The authors conducted an evaluation of the nuclear reaction yield of protons, α-particles and 1-neutrons in material equivalent to human tissues in proton therapy using simulations with GEANT 4. In addition, the authors showed that the enhancement of nuclear reaction products by 5 radiosensitizer nanoparticles turns out to be negligible.

Reviewer's opinion, the article fits well into the subject of the journal and after language correction it can be published in  International Journal of Molecular Sciences.

Moderate editing of English language required.

Author Response

Dear referee,

Thank you for reviewing our paper and giving a positive response. 
Your remarks were very valuable for us. We have carefully read the paper 
few more times and  made some language corrections.

Kind regards,
Martin, Maxim, Vladimir

Round 2

Reviewer 2 Report

The authors have answered the questions and points raised in my first review of the manuscript. The work can be published in this form.